# Invasive potential of tropical fruit flies in temperate regions under climate change

Andrew Paul Gutierrez [1,2✉], Luigi Ponti [1,3✉], Markus Neteler [4], David Maxwell Suckling[5,6] &
José Ricardo Cure[1,7]

Tropical fruit flies are considered among the most economically important invasive species detected in temperate areas of the United States and the European Union. Detections often trigger quarantine and eradication programs that are conducted without a holistic understanding of the threat posed. Weather-driven physiologically-based demographic models are used to estimate the geographic range, relative abundance, and threat posed by four tropical tephritid fruit flies (Mediterranean fruit fly, melon fly, oriental fruit fly, and Mexican fruit fly) in North and Central America, and the European-Mediterranean region under extant and climate change weather (RCP8.5 and A1B scenarios). Most temperate areas under tropical fruit fly propagule pressure have not been suitable for establishment, but suitability is predicted to increase in some areas with climate change. To meet this ongoing challenge, investments are needed to collect sound biological data to develop mechanistic models to predict the geographic range and relative abundance of these and other invasive species, and to put eradication policies on a scientific basis.

[1] Center for the Analysis of Sustainable Agricultural Systems (www.casasglobal.org), Kensington, CA, USA. [2] Division of Ecosystem Science, College of Natural Resources, University of California, Berkeley, CA, USA. [3] Agenzia nazionale per le nuove tecnologie, l'energia e lo sviluppo economico sostenibile (ENEA), Centro Ricerche Casaccia, Roma, Italy. [4] mundialis GmbH & Co. KG, Bonn, Germany. [5] The New Zealand Institute for Plant and Food Research Ltd., Christchurch, New Zealand. [6] School of Biological Sciences, The University of Auckland, Auckland, New Zealand. [7] Facultad de Ciencias Básicas y Aplicadas, Universidad Militar Nueva Granada, Bogotá, Colombia. ✉email: CASAS.Global@berkeley.edu; luigi.ponti@enea.it

Natural and agricultural systems worldwide are increasingly threatened by mismanagement, overharvesting, by climate and global change, and by an increasing incidence of invasive species, all of which are related to human population growth and activity. Ongoing concern about the potential of tropical fruit flies to invade US agriculture has led to a substantial development of surveillance and eradication infrastructure starting in the 1930s.

Among the most important invasive insect species are fruit flies of the family Tephritidae, with the most economically important being from four genera: *Ceratitis*, *Anastrepha*, *Bactrocera*, and *Rhagoletis*. Tephritid flies have had considerable success in invading new regions of the world, especially tropical and sub-tropical areas. Their success has been attributed to their wide host range and developmental response and tolerance to environmental variables through phenotypic plasticity and genetic adaptation[1–7]. Several species of tropical fruit flies have been detected in temperate areas of California and other states (California Department of Agriculture (CDFA) data[8]), and in the European Union (EU)[9,10]. Their potential threat has been couched in economic terms; to say the $25 billion California fruit and vegetable industry. On average, a single eradication campaign against tropical fruit flies in the USA is estimated to cost approximately US $32 million[8] and up to US$100 million as occurred for medfly in California during 1980–1981[11]. These eradication programs were conducted without a holistic understanding of the threat posed or of the potential geographic range of the tropical species.

Weather-driven physiologically-based demographic models (PBDMs)[12] of fruit fly biology and dynamics are used to predict the prospective geographic range and relative abundance (i.e., invasive potential) of four tropical species in North and Central America, and in the European-Mediterranean region under extant weather and climate change. The target species include the Mediterranean fruit fly (*Ceratitis capitata;* medfly) from East Africa, melon fly (*Bactrocera cucurbitae*) native to India, oriental fruit fly (*Bactrocera dorsalis*) from Asia, and the Mexican fruit fly (*Anastrepha ludens*; mexfly) native to Mexico and Central America. Medfly is established in the Mediterranean Basin and in Central America, oriental and melon fly are widely distributed in tropical regions of the Eastern Hemisphere, and mexfly occurs in Mexico and Central America. Of the four species, only the Mexican fruit fly is not established in Hawaii.

In this paper, we address two related issues: what is the potential of the four tropical fruit flies to invade temperate areas, and what is their prospective geographic range under extant and climate change weather. We note that from the perspective of an ectotherm species, climate change is another weather pattern that may or may not enable them to persist and to invade new areas.

Cold weather restricts the northward limits of tropical fruit flies in temperate regions, but high temperatures and low relative humidity may also limit reproduction, survival, and permanence in seemingly favorable areas. Weather-driven PBDMs allow examination of the dynamics of a species at any location, across time, geographic space, and climate change, enabling identification of areas and times of favorability.

However, in developing ecological models, we must keep in mind the metaphor of the Precision Corollary of Murphy's Law: what is measured with a micrometer, may be marked with chalk, and then cut with an axe—i.e., ecological models must be viewed as approximations of the biology[13]. Given this homily, we note that the underlying biology of tropical fruit fly species is similar, but their responses to weather and hosts differ—differences that determine their potential geographic distribution and abundance. Briefly, fruit fly egg and larval stages are found in host tissues, mature larvae usually leave the hosts and form pupae in soil, and emerging adults are free living and have the ability to seek more favorable conditions in the local environs. Temperature is the major driving variable, with relative humidity also affecting survival and reproductions in adults. Dormancy (diapause) does not occur in any of the four species, but adults may be long-lived when reproduction is reduced or ceases due to adverse temperature and relative humidity, or when hosts are unavailable. Models for host plant dynamics are not available (Table 1), and hence in our model the flies are assumed to reproduce as weather conditions allow. In PBDMs, the weather-driven dynamics of species are captured by biodemographic functions (BDFs, Fig. 1) that characterize the biology, and are used to estimate the effects of temperatures on developmental rates and mortality of all stages, and the effects of temperature and relative humidity on reproduction[14]. The simple mathematics of PBDMs and BDFs, and the biological data underpinning them are reviewed in the methods section, while additional discussion of the biology of the species and maps of their geographic distribution and relative abundance are summarized as Supplementary Figs. S1–S10.

Parameterization of BDFs is best with data from studies designed to develop PBDMs, but such data are rare, including in well-funded sterile insect technique (SIT) eradication programs for tropical fruit flies, and the new world screwworm (*Cochliomyia hominivorax*)[14] costing hundreds of millions of dollars. Here, we used data from a diverse literature to parameterize the BDFs, noting that the data required checking for internal consistency (outliers), and where possible comparisons to other data sets for the same factor, before the BDF parameters could be estimated/deduced with relative confidence. The relative adequacy of the data used to parameterize the

**Table 1 Relative adequacy of data used to parameterize the BDFs for five invasive fruit fly species: ª host range is polyphagous, or host specific, ᵇ adequacy of data: sufficient (+), marginal (-), insufficient (o).**

| Factor/species | Olive fly | Mediterranean fruit fly | Oriental fruit fly | Melon fly | Mexican fruit fly |
|---|---|---|---|---|---|
| Climate type | Subtropical | Subtropical | Tropical | Tropical | Subtropical |
| Origins | Eastern Africa | Eastern Africa | Asia | India | Mexico-Central America |
| Host specificity[a] | Specific | Polyphagous | Polyphagous | Polyphagous | Polyphagous |
| Host model[b] | + | o | o | o | o |
| Functional response | + | + | + | + | + |
| Developmental rate | + | + | + | + | + |
| Fecundity | + | + | + | + | + |
| Temp. ovip. scalar | + | + | + | – | – |
| RH ovip. scalar | + | + | – | – | – |
| Temp. mortality | + | + | + | – | + |

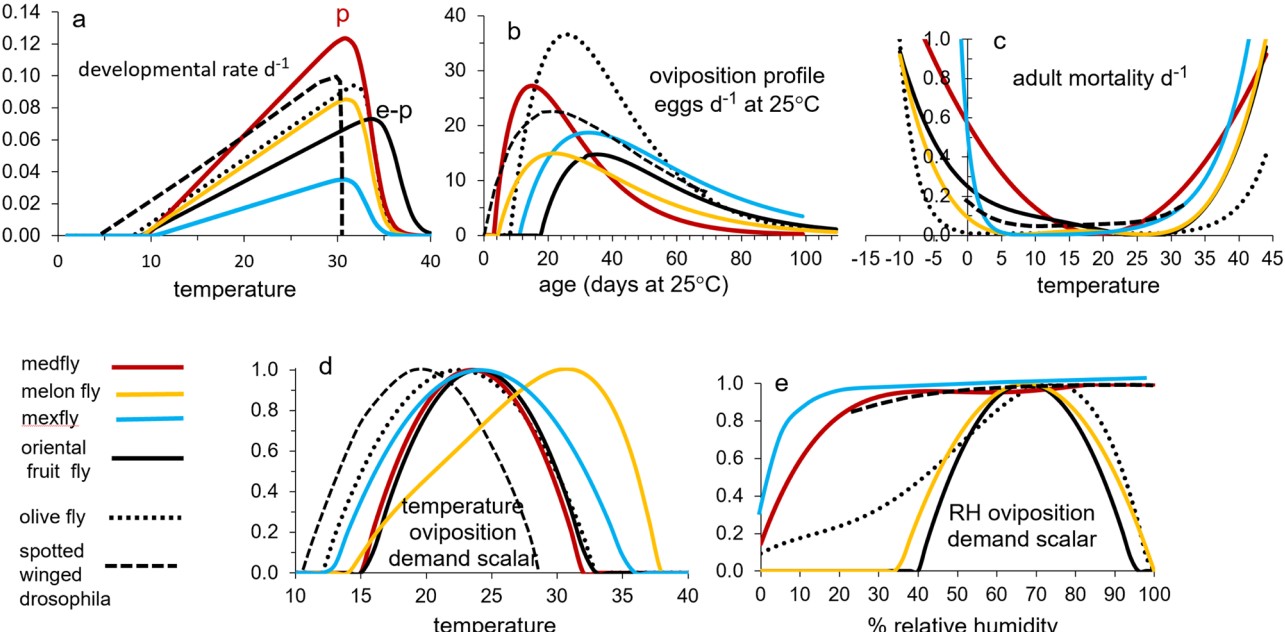

**Fig. 1 A summary of the biodemographic functions (BDFs) for six fruit flies. a** Rates of development on temperature (symbols e-p are egg-pupal stages for melon fly and oriental fruit fly, p for medfly pupal stage, and all others are e-l for egg-larval stages), **b** the age specific oviposition profiles, **c** temperature dependent mortality rates, and **d**, **e** are temperature ($\phi(T)$) and relative humidity ($\phi(RH)$) scalars for adult reproduction. The the BDFs and paramenters for the four fruit flies in this study are presented in Figs. 5–8 and Table 2, and the data are reported in Supplementary EXCEL files S1–S4. The BDFs and parameters for olive fly and spotted wing dosophila are reported in Supplementary Table S1.

BDFs of the four species are summarized in Table 1, the parameterized BDFs are reported in Table 2, and the data used are summarized in Supplementary Excel files S1–S4. For some species, data were not available on the effects of temperatures and RH on reproduction, and the BDFs were inferred from cited reports. Included for comparative purposes are BDFs for the invasive host specific olive fly (*Bactrocera oleae*) that is well established where olive (*Olea europaea*) is cultivated in our study areas, and for the Asian spotted-wing drosophila (*Drosophila suzukii*, family Drosophilidae) that has invaded large areas of North America and the Palearctic[15]. The BDFs for olive fly and *D. suzukii* are summarized in Fig. 1 and Supplementary Table S1.

The BDFs are implemented in age-structured distributed-maturation time population dynamics models (Supplementary Fig. S11)[12] that are driven by daily weather over multiple years. The model computes the daily age structured dynamics of each species in >30,000 lattice cells across the vast geographic areas of North and Central America, and the European-Mediterranean region (see "Methods" section), but only the average cumulative number of pupae per year is used as a metric of relative favorability in each lattice cell. The pupal data are mapped and analyzed using GRASS GIS[16]. Maps for sub regions can be developed from the data to provide finer grain detail.

## Results

**North and Central America**. The simulation data (average pupae) for the prospective distributions and relative abundance of the four fruit flies for the period 1980–1990 in North and Central America are mapped in Fig. 2. In all figures, normalized pupal densities may be viewed as qualitative indices of favorability ($0 \le FI \le 1$), with $FI \le 0.5$ indicating low decreasing levels of favorability, and $0.5 > FI \le 1$ indicating increasing favorability and potential for establishment. More detailed maps are provided as Supplementary Figs. S1–S10.

*Medfly*. The prospective distribution of medfly is largely restricted to tropical regions of Mexico and Central America (Fig. 2a). The high elevation areas of Mexico are predicted unfavorable as are the upper reaches of the NW deserts regions and Baja California. Small areas of coastal southern California and southern Florida have favorability indices of ~0.5 suggesting marginal favorability. Eradication efforts are active in Central America, Mexico, and the USA when the fly is detected, and the areas are posited free of medfly (Supplementary Fig. S1a, b).

*Melon fly*. Prospectively, the tropical areas of Mexico and Central America are highly suitable for melon fly, with favorability decreasing northward along the eastern coast to Florida where favorability again increases (Fig. 2b). The FI = 0.5 isoline indicated in white, suggests the upper geographic limits of marginal favorability (Supplementary Fig. S2a–d). Melon fly commonly attacks melon, cucumber, and tomato, and as a backdrop for the prospective distribution of the fly, the considerably larger area of tomato cultivation in North and Central America is mapped in Supplementary Fig. S2g.

*Oriental fruit fly*. The prospective range of oriental fruit fly is similar to that of melon fly, but with a smaller range and 2.5-fold greater pupal densities in favorable areas (Fig. 2c). The desert regions of North America are unfavorable, with coastal southern California having marginal favorability. Large numbers of detections have occurred in coastal southern California since 1965, and hence a detailed analysis on the effects of weather on the fly's lack of establishment is reported in Supplementary Figs. S3–S5.

*Mexican fruit fly*. The prospective distribution of the native mexfly is similar to, but more extensive than that of the exotic

**Table 2 Summary of biodemographic functions (BDFs) and parameters of four invasive tropical tephritid fruit flies from Figures 5-8.**

| BDF/ parameters | Mediterranean fruit fly | melon fly | Mexican fruit fly | oriental fruit fly |
|---|---|---|---|---|
| del-time ($\Delta$) egg-larvae pupae adults quiesent adults | 129dd > 10.345°C<br>165dd > 9.5°C<br>772dd > 9.5°C<br>1050dd > 9.5°C | 104dd > 7.95°<br>164dd > 7.95 °C<br>1197dd >7.95°C<br>1197dd >7.95°C | 258dd > 10.3°C<br>264dd > 10.3 °C<br>1000dd >10.3°C<br>1000dd >10.3°C | 136dd > 8.87 °C<br>177dd >8.87°C<br>1050dd >8.87°C<br>1532dd >8.87°C |
| $^{E\text{-}L}R(T)$<br>$^{A}R(T)$, $^{A}\Delta_x(T)$ | $0.031(T-10.35)/(1+2.75^{T-33.5})$<br>$0.0059(T-9.5)/(1+4.8^{T-33.7})$ | $0.00975(T-7.95)/(1+4^{T-33.5})$ for E-L<br>$^{A}\Delta_x(T) = {}^{P}\Delta_x(T) = {}^{E\text{-}L}\Delta_x(T)$ | $0.0018(T-10.3)/(1+4^{T-34})$ for E-P<br>$^{A}\Delta_x(T) = {}^{E\text{-}P}\Delta_x(T)$ | $0.00305(T-8.87)/(1+3.25^{T-36.5})$ for E-P<br>$^{A}\Delta_x(T) = {}^{E\text{-}P}\Delta_x(T)$ |
| $^{E\text{-}L}k$, $^{P}k$, $^{A}k$ | 25, 40, 50 | 25, 40, 70 | 25, 40, 68 | 25, 25, 71 |
| $f(x)$ age in days at 25°C | if $x \geq 3$ then<br>$6.25(x - 3)/(1.088^{(x-3)})$ | if $x \geq 4$ then<br>$2.55(x - 4)/(1.065^{(x-4)})$ | if $x \geq 11$ then<br>$2.355(x - 11)/(1.0475^{(x-11)})$ | if $x \geq 16$ then<br>$2.3(x - 16)/(1.065^{(x-17.5)})$ |
| $0 < \phi_T \leq 1$ | $F(T, 15.0, 32.0)$ | $0.0034 (T-13.3)/ (1+1.85^{T-34.2})$ | $F(T, 12.78, 32.5)$ | $0.0006 T^3 -0.0567 T^2 + 1.6652T -14.603$ |
| $0 < \phi_{RH} \leq 1$ | $1 - 0.000000063RH^4 + 0.000016229RH^3 -0.001487393RH^2 + 0.057917778RH +0.15$ | $F(RH, 35, 100)$ | $1 - 0.0000000635RH^4 + 0.0000162RH^3 - 0.0014874RH^2 + 0.0579178RH + 0.145$ | $F(RH, 40, 95)$ |
| $0< {}^{E\text{-}L}\mu\,(T) \leq 1$ | $0.000005T^4 - 0.000444T^3 + 0.0150147T^2 - 0.217793T + 1.16$ | $0.00000232T^4 -0.00015536T^3 + 0.003497847T^2 - 0.02999476T + 0.0909$ | $\begin{cases} \text{if } T<10.3°C \text{ then} \\ 0.518\,e^{-0.690312T} \\ \text{else} \\ 0.00033\,e^{0.1932889T} \end{cases}$ | $0.00000187T^4 - 0.0001116T^3 + 0.0024931T^2 - 0.0308056T + 0.2486$ |
| $0< {}^{P\text{-}A}\mu\,(T) \leq 1$ | $0.000487T^2 -0.018705T + 0.1846$ | $^{P\text{-}A}\mu = {}^{E\text{-}L}\mu$ | $^{P\text{-}A}\mu = {}^{E\text{-}L}\mu$ | $^{A}\mu = {}^{E\text{-}P}\mu$ |

$T$ = mean temperature, $RH$ = mean % relative humidity, $k$ = stage specific Erlang parameter in eqn. 1. Scripts **E, L, P, A** = egg, larval, pupal, and adult stages, $f(x)$ = *per capita* reproductive rate per day on age ($x$) at 25°C, $^{stage}\Delta$ = developmental time constant in dd above the lower threshold for a stage, $^{stage}\Delta_x$ = (stage developmental time multiplied by stage developmental rate), $0 \leq F(T$ or $RH$, **min, max**) $\leq 1$ is a symmetrical function between min and max values equal 1 at (max + min)/2). $\phi_{RH}$ and $\phi_T$ are scalars for reproduction, and $\mu_{stage}$ is the temperature-dependent stage mortality rate. Fits for all BDFs had $R^2 > 0.85$.

medfly, with mexfly pupal densities in favorable areas being 40% higher. Coastal southern California is only marginally favorable (FI < 0.5) (Fig. 2d and Supplementary Fig. S6a, b).

**The European-Mediterranean region**. The prospective distribution of tropical fruit flies in the Palearctic region (Fig. 3) is limited northward by cold weather and in the southern reaches of Saharan North Africa by hot weather, low relative humidity, and lack of hosts. Medfly, melon fly and oriental fruit fly are well established throughout tropical sub Saharan Africa[17], and medfly is established in some areas of the European-Mediterranean region.

*Mediterranean fruit fly*. The prospective distribution of the fly is mapped in Fig. 3a, and is projected to be most abundant in the Nile delta of Egypt, with lower densities in coastal and near coastal Morocco and Mediterranean North Africa, and the Levant. Lower favorability is predicted for Spain, near coastal areas of southern France, Italy and the coast of Syria and Lebanon. Predicted highest densities in favorable areas of the Mediterranean Basin are ~20% higher than in Mexico-Central America. The predicted geographic distribution of the fly compares favorably with those of correlative methods CLIMEX[18] and Principal Components Analysis[19] (Supplementary Fig. S1e, f).

*Melon fly*. The simulation data for melon fly are mapped within the much larger distribution of tomato (Supplementary Fig. S2h). Most areas of the European-Mediterranean region have low favorability for melon fly (FI < 0.25), with the most favorable areas restricted to areas along the north African coast (FI ~0.5), coastal areas of Israel (FI ~0.6), with highest favorability predicted in the Nile Delta of Egypt (Fig. 3b and Supplemental Fig. S2e, f). We note that maximum pupal populations in the region are <70% those predicted for favorable areas of Mexico and Central America (Figs. 2b vs. 3b).

*Oriental fruit fly*. The prospective distribution of oriental fruit fly is restricted primarily to the Nile Delta, and near coastal regions of North Africa, southwestern Spain, and Israel. The highest densities are predicted for the Nile delta that are ~60% those in Mexico-Central America (Figs. 2c vs. 3c). FI < 0.5 is the rule for most of Europe and Turkey. Removing the effects of relative humidity on reproduction does not greatly increase population levels or the geographic range (see Supplementary Fig. S5d vs. S5e).

*Mexican fruit fly*. Mexfly has not been detected in the Palearctic. Its prospective distribution is similar, though considerably less than that of medfly (Fig. 3a vs. 3d). The highest potential densities are predicted in Morocco, coastal North Africa and southern Portugal and Spain, small areas of Sicily and south Italy, Crete, and part of the Levant (Supplemental Figs. S6c, d). The highest predicted population densities are ~17% less than in its native areas of Mexico and Central America.

**Effects of climate change on prospective distributions**. The prospective distributions of the four fruit flies in North and Central America and the European-Mediterranean region under climate change are shown in Fig. 4, with greater detail presented in Supplementary Figs. S7–S10.

*Mediterranean fruit fly*. The prospective distribution of medfly in North and Central America is predicted to increase with population levels doubling in highly favorable areas (Fig. 4a and Supplemental Fig. S7a vs. Fig. 2a). The coastal plain of coastal southern California is predicted to become more favorable (FI ~0.7, Fig. 4a, inset in Fig. S7a).

The distribution of medfly in the European-Mediterranean region is projected to decrease in North Africa, but to increase in the Levant, Cyprus, Crete, and southern Greece (Supplemental Fig. S7b and Figs. 4a vs. 3a). Projected favorability is marginal to unfavorable (FI < 0.5) in most areas because of

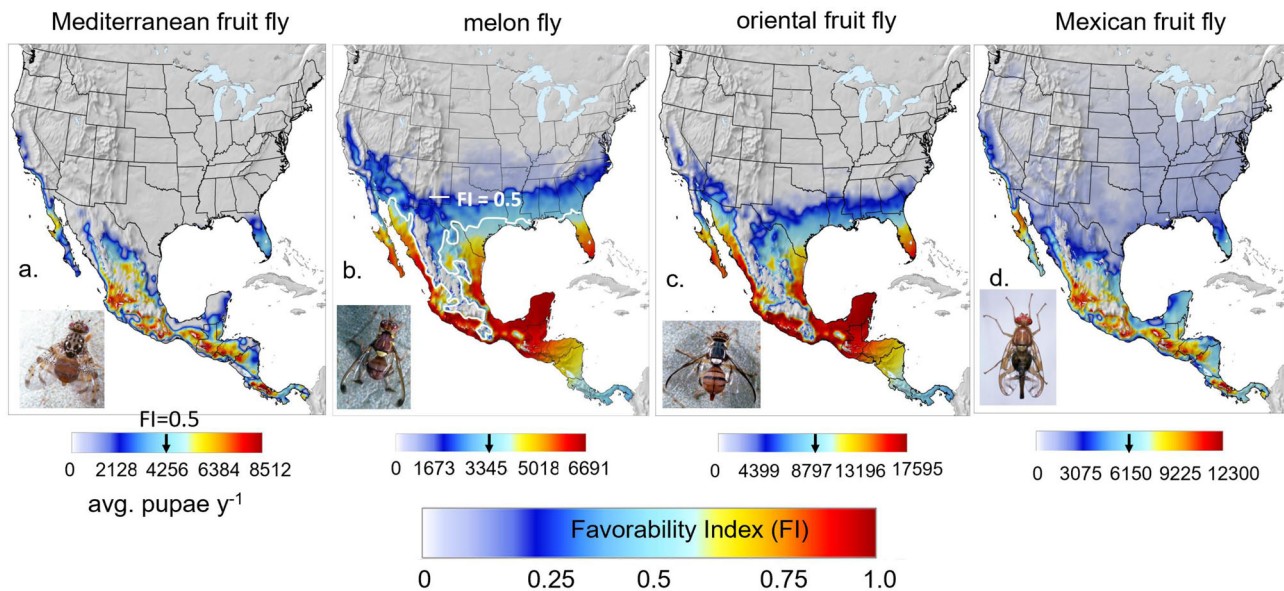

**Fig. 2 Prospective average distribution and relative abundance (average annual sum of pupae) in North and Central America during 1980–1990. a** Mediterranean fruit fly, **b** melon fly with the FI = 0.5 isocline indicated in white, **c** oriental fruit fly, and **d** Mexican fruit fly (detailed maps in Supplementary Figs. S1, S2, S5, S6). Clips of the tropical fruit flies are by Jack Kelly Clark, University of California Statewide IPM Program.

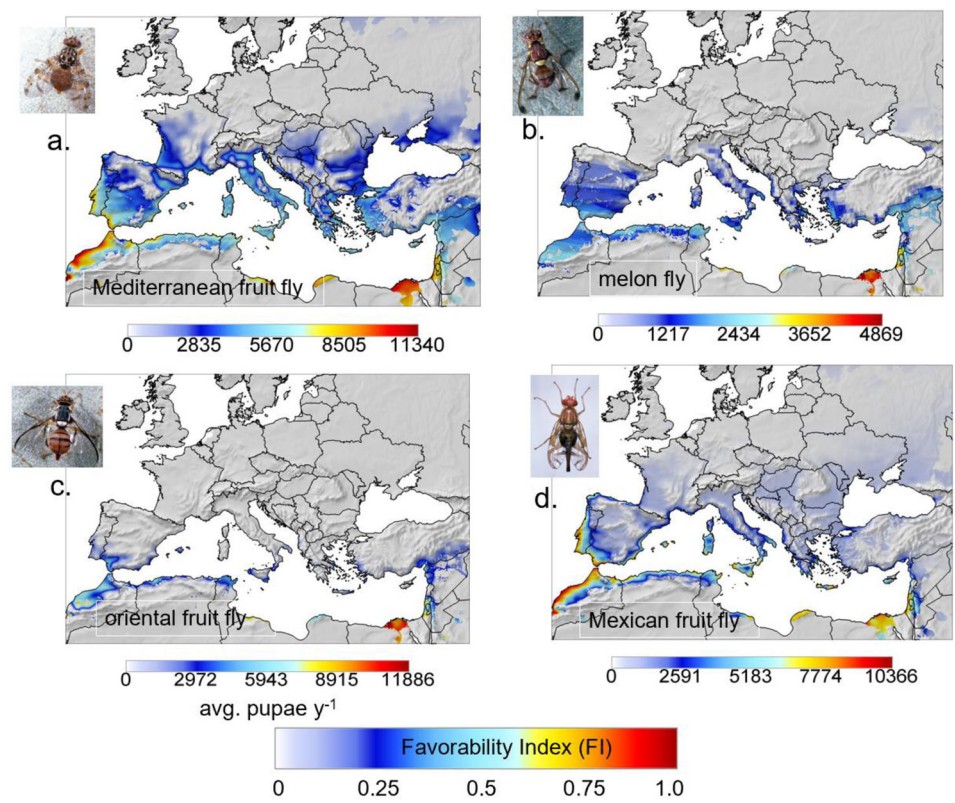

**Fig. 3 Prospective average distribution and relative abundance (average annual sum of pupae) in the European-Mediterranean region during 1980–1990. a** Mediterranean fruit fly, **b** melon fly, **c** oriental fruit fly, and **d** Mexican fruit fly (detailed maps in Supplementary Figs. S1, S2, S5, S6). Clips of the tropical fruit flies are by Jack Kelly Clark, University of California Statewide IPM Program.

increased summer temperatures in southern areas, and continued cold winter weather in northern areas. High population levels of pupae are still predicted in favorable areas such as the Nile Delta, but generally, conditions in the European-Mediterranean region are predicted to become increasingly marginal to unfavorable.

*Melon fly*. The prospective maximum abundance of the melon fly in favorable areas of Mexico and Central America decreases slightly, with the area of favorability expected to increase along the eastern coast of Mexico, Central America, the US Gulf states, and Florida (Fig. 4b). The shift in mean favorability is seen in the northward shift of the FI = 0.5 isocline (Supplemental Fig. S8a

and Figs. 4b vs. 2b). Increases in favorability occur in western Mexico into coastal southern California, with some increase in marginal favorability in the Central Valley of California.

Most of the European-Mediterranean region remains unfavorable for melon fly with small decreases in the most favorable areas such as the Nile Delta (Supplemental Fig. S8b and Figs. 4b vs. 3b). This could be due to the projected decline in rainfall of up to 40% in some areas of the Mediterranean Basin[20], and associated reduced relative humidity that affects melon fly reproduction.

*Oriental fruit fly*. A slight expansion of range is predicted in North America, with the greatest increases in favorability projected for Central America, with small increases in range in western Mexico and Baja California. Modest increases in favorability are predicted in the southern USA (Supplemental Fig. S9b and Figs. 4c vs. 2c).

In the Euro-Mediterranean region, maximum population size remains relatively unchanged, with the Nile Delta remaining favorable, though populations decrease about 10%. The prospective geographic range of favorability increases slightly in Morocco and near coastal areas of the Levant. Increases, but still low favorability, occur in other areas (e.g., Spain, Supplemental Fig. S9b and Figs. 4c vs. 3c).

*Mexican fruit fly*. The endemic favorable range of mexfly is expected to decrease in the Yucatan, and north central and east coastal Mexico, but favorability will increase in Baja California and along coastal California (Supplemental Fig. S10a, Figs. 4d vs. 2d).

Prospective favorability is expected to decrease in North Africa, coastal Portugal and Spain and the Levant (Supplemental Fig. S10b and Figs. 4d vs. 3d).

## Discussion

Predicting the potential invasive range of invasive species is essential to informing quarantine and control measures, and developing appropriate models to do this is critical. Among the assessment alternatives used are de facto standard correlative methods, inferences from detection records, and intuitive verbal projections. Correlative methods (e.g., species distribution models (SDMs) and ecological niche models (ENMs)) use species occurrence data to find correlates in aggregate weather to circumscribe the climatic envelope of the species. SDMs and ENMs results are often used to extrapolate the potential geographic range in uninvaded areas under extant weather, and with climate change. SDM-ENM approaches assume the species distribution records are valid, but the potential geographic range of a species may be limited by undiscovered abiotic and biotic factors (e.g., intrinsic plasticity, natural enemies, and competitors)[21], making the transferability of SDM-ENM results uncertain[22]. For example, neither the correlative MaxEnt or boosted regression tree models were able to predict the high climatic suitability of the host-specific olive fly in its entire invaded range[23]. Further, a recent well-documented failure of the SDM-ENM approach are two early correlative CLIMEX analyses of the invasive potential of the South American tomato pinworm (*Tuta absoluta*) in the European-Mediterranean region[24]. The CLIMEX analyses failed to predict the invasion of *T. absoluta* in colder areas because the species distribution records used to fit the model were from tropical regions, and did not capture its moderate tolerance to cold weather. A later CLIMEX analysis included records from the invaded temperate region, and gave after the fact predictions similar to a PBDM developed without reference to detection records for the pest[24].

Further, there is a need to move beyond verbal arguments of invasive potential that ignore objective criteria or are based on statistical inferences from detection data that ignore weather effects[25]. Based on statistical inferences, it has been claimed that at least five to nine invasive fruit fly species, including Mediterranean fruit fly, Mexican fruit fly, oriental fruit fly, melon fly, and possibly the peach fruit fly (*Bactrocera zonata*) and the guava fruit fly (*B. correcta*) are established in California; albeit "at undetectable levels"[8]. Such claims cannot be verified nor falsified, and were made without considering the role of weather on the potential for establishment and of the geographic range of the flies[26,27]. Such claims ignored microsatellite and mitochondrial DNA evidence of multiple introductions for medfly and oriental fruit fly[28–30], and the valid notion that if the fruit flies are established in California and the climate is favorable, then population growth would occur and the flies would be detectable[27] (Supplementary Figs. S3–S5 and associated analysis for oriental fruit fly). None of the four species considered in our study are established in California (Dr. Kyle Beucke, Primary State Entomologist, California Department of Food and Agriculture (CDFA), personal communication), but this may change with climate warming[26].

Some posit that quarantine and eradication measures by state and federal agencies prevented the invasion by tropical fruit flies in the USA[27], and yet several invasive species in several taxa have invaded California, and quickly established. Among them are the subtropical olive fly that entered California in 1998 and quickly invaded all areas of olive cultivation[27,31,32]. Similarly, the Asian spotted wing drosophila became widely distributed in North America and the Palearctic region[33]. Compared to the four tropical species in our study (Fig. 1), these two species have lower thermal thresholds for development and reproduction, and are relatively cold tolerant (Supplementary Table S1). The BDFs that characterize the biology of these two fruit flies in a PBDM/GIS context yielded different prospective geographic distributions in North America and the European-Mediterranean region that accord with their known geographic distribution[14,31,33,34].

For invasive species to successfully invade new areas, their life history traits and biology (physiology and behavior) must enable them to survive the time varying patterns of weather and biotic constraints. Recognition of such constraints has long been part of the theory and practice of ecology[35] and biological control[36], but the question of how to model the complexity of species dynamics and their invasiveness as driven by weather in natural and agricultural ecosystems has been elusive. But, is it even possible as Palladino's[37] fundamental question asks: *Is nature idiosyncratic?* Some sanguinely appraise that it is too complicated[38], that there is a need for studies that document variation in all of the parameters[21], and that "...*ecological modeling, is* [correctly]... more a heuristic tool than a surrogate for reality"[13]. And yet we need to develop "... models based on understanding the processes that result in a system behaving the way it does,... remaining valid indefinitely"[39]. To do this, we must adopt a science vision of system ecology with reliable rules (laws) of nature common to all ecosystems and species that include mathematical descriptions of population dynamics and trophic interactions, and conforms to the laws of thermodynamics[40]. The approach must be holistic, enabling the development of models of a species' biotic responses to environmental variables—models to assess their climatic-geographic limits and time-place dynamics. Although not couched as physics, this is what the PBDM approach attempts.

And yet, species with wide geographic distributions may exhibit phylogeographic structure, with lineages potentially

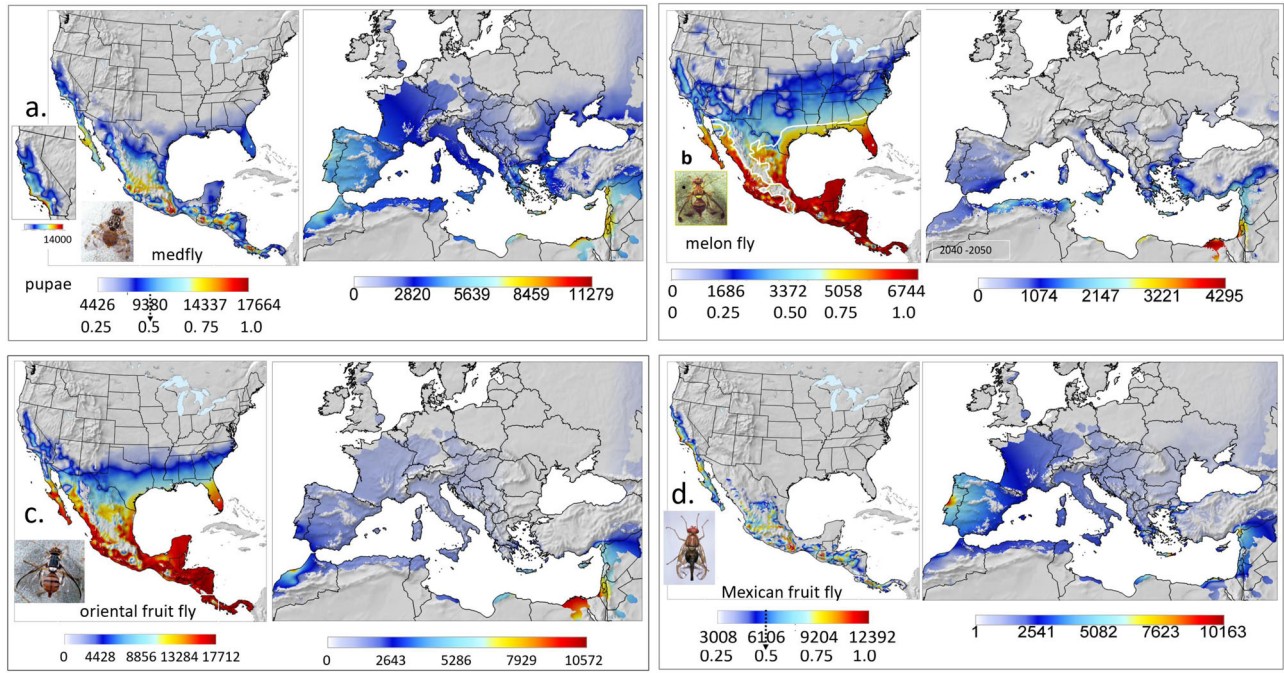

**Fig. 4 Prospective average distribution and relative abundance (average annual sum of pupae) under climate change. a** Mediterranean fruit fly, **b** melon fly with the FI = 0.5 isoline indicated in white, **c** oriental fruit fly and **d** Mexican fruit fly in North and Central America (2055–2065), and in the European-Mediterranean region (2040–2050) (detailed maps in Supplementary Figs. S7–S10). Clips of the tropical fruit flies are by Jack Kelly Clark, University of California Statewide IPM Program.

adapted to different biotic and abiotic conditions, and hence the success of an invasion may depend on the intraspecific identity of the introduced population[23,41]. That biotypes of species exist and respond differently to hosts and to climate has also long been an accepted concept in the field of biological control[42]. Lineages of the same species may display different life history traits related to adaptations to local climate[43–45], including differing tolerance to cold (e.g., *B. dorsalis*[46]) and may have different host preferences[47]. While genomic differences are often reported for different populations of a species, they usually provide no indication as to whether the populations have different climatic requirements. An excellent study[48] found four genomic distinct populations of *A. ludens* (western and eastern Mexico, and northern and southern Central America) that would be useful in determining origins of invasive propagules, but not their invasive potential. These and other issues would appear to question the validity of species-based pest risk assessments, that if widely applicable, intra-specific variation would place an impossible data burden in estimating climatic limits of suitability for invasive species[23].

However, the existence of a limited scope [capacity] for further adaptation to increasing heat resistance in a rapidly warming planet, coupled with the narrower thermal safety margins tropical ectothermic species display, would appear to limit thermal tolerance evolution on a short evolutionary time scale[49]. Moreover, the survival and thermal requirements of tropical, subtropical, and temperate populations of *C. capitata* in Brazil were found to be similar, demonstrating the species' capacity to adapt to different climatic conditions based on the same biological attributes (i.e., plasticity)[50], and further, the minimum developmental thresholds for medfly larvae from different areas are remarkable similar (~9–10 °C, Fig. 5)[43,50,51]. Reported differences in thermal thresholds and other parameters may be real[23], but may also be due to experimental methods, observation intervals, accuracy of

recorded temperatures, and other factors. Laboratory studies using artificial diet found that medfly populations from several regions exhibited differences in vital rates, suggesting that geographically isolated medfly populations may have different invasive potential[6,7], or the results may have been due to different responses to the diet used in the study.

The literature on tropical fruit flies consists of a panoply of laboratory studies, of age-specific life tables, field ecology, reproductive maturation, host preferences, thermal responses, mating behavior, temperature treatments designed to develop quarantine measures, and academic topics such as aging in fruit flies—of data gathered for disparate purposes. However, the process of comparing data was facilitated by the fact that many studies of the four tropical fruit flies in our study were conducted at ~25 °C. More important, the within-species parameters estimated from the literature for the four fruit flies were remarkably consistent, albeit incomplete. The major issue of how to come to grips with this cacophony of partial information was facilitated by the PBDM/BDF approach.

Firstly, nature is not idiosyncratic, as the underpinning bioeconomic rules (BDFs) are similar across species (Fig. 1)[14]. Further, we propose that general demand-driven PBDMs having sound biological, physiological[12], and bioeconomic underpinnings[52] can capture the dominant features of the biology of species (and of geographic variants), and can be used to assess their relative abundance and geographic distribution under extant and climate change weather[24]. However, estimating the BDFs from the literature for the four tropical fruit flies was vexing and arduous, and revealed important data gaps—especially at the extremes of temperature and for RH (Tables 1, 2 and Figs. 5–8, and Supplementary Excel files S1–4). And yet, the prospective geographic distributions and relative favorability in North and Central America and in the European-Mediterranean region predicted by the PBDMs accord with available correlative species

distribution studies based on species occurrence records for medfly and oriental fruit fly (Supplementary Figs. S1e, f and S5g, h respectively). This occurred despite model simplifications and assumptions, a paucity of available data to parameterize fully the biodemographic functions (Table 1), and the imprecision of observed and climate model weather data used to drive the PBDMs. Similar good predictions have accrued for other species[14].

Hundreds of millions of dollars have been spent on sterile insect technology (SIT) eradication programs against the new world screwworm[53,54] (and the tropical fruit flies in this text), and yet, the requisite holistic data on their biology have not been published. This likely occurred because the programs lacked an appropriate modeling framework able to integrate the data and project the results regionally. And hence, researchers focused attention on aspects that had immediate practical application or simply appealed to scientific curiosity. Yet despite data limitations, the PBDM for screwworm showed clearly that screwworm had limited capacity for permanence in the southern USA, explained when and why outbreak occurred in Texas during the eradication period, and predicted its permanent range expansion under climate change[53,54]. PBDMs are able to capture effects of unfavorable short-term weather (e.g., the severe winter weather as occurred in the SE USA during February, 2021) that can limit the permanence of screwworm (and other species) in an area, and the results can be used in real time to inform control and management strategies.

In California during 1960–2010, when controversy concerning the potential invasion by tropical fruit fly species was rampant, the state was not particularly favorable for the four species of tropical fruit flies in our study. The threat of their establishment under 1980–1990 weather was marginal in coastal southern California and southern Florida. However, with climate change and unrelenting introductions, invasions of larger areas of near coastal California by tropical fruit flies could change as condition becomes more tropical and favorable (e.g., for medfly and mexfly specifically, Supplementary Figs. S7, S10 vs. Fig. 2). The effects of climate change in the European-Mediterranean region are more complex, but the PBDMs showed where the potential range of the tropical fruit flies may increase or contract. The capacity to predict these changes is critical in dealing with ongoing invasions by these and other pests, as recent detections of oriental fruit fly in Europe show[55,56].

The magnitude of the invasive species problems calls for coordinated and focused efforts in developing sound holistic biological data on economically important species by well-funded agencies—data to parameterize mechanistic physiologically based models (including PBDMs) to help guide costly control and eradication policy[14,57,58]. Compared to the large sums spent on control and eradication efforts on these and other pest species, the costs and effort of gathering the appropriate biological data to develop well parameterized weather-driven mechanistic models for them would be pitifully small, and would yield considerable public benefit in evaluating their invasive potential under current weather and climate change. While the social and technical components of eradication are challenging, improved models can provide a practical method for assessing invasion and economic risk, reducing the need for verbal proclamations, and of judgment based on experience often acquired based on prior bad judgment (cf. American humorist Will Rogers).

## Methods

**Population dynamics model.** PBDMs are physiologically-based time-varying life tables[59,60], the theoretical basis of which has been reviewed[61–63]. It is commonly supposed that PBDMs have large numbers of parameters making them difficult to develop[64], but as demonstrated here and in numerous prior studies, this is not the

case (Table 2 and Supplemental Table S1), and further, the same underlying model(s) can be used across species and trophic levels[14].

Only an overview of the time-invariant distributed-maturation time demographic model used in our study[65] is presented here, and as Supplementary information. N.B. Other demographic models could also be used[66,67]. Tropical fruit flies have egg-larval, pupal, and adult life stages (left superscript s), and the same discrete dynamics model is used for all of them. The model for the $i$th age class of a life stage with $i = 1, 2,…,^sk$ age classes (Supplementary Fig. S11a) is Eq. 1[12,68]. The model can be viewed as $^sk$ dynamics equations for each stage. The forcing variable is temperature ($T$), with time ($t$) being a day (d) that from the perspective of ectotherm stage of variable length in physiological time units (i.e., $^s\Delta_x(T(t))$ in degree days (dd)), or proportional develoment $^sR(T(t))$) that may differ for each stage having different mean developmental times $^s\Delta$. The state variable $^sN_i(t)$ is the density of the $i$th age class, and $^s\mu_i(t)$ is the proportional age specific net loss rate due to temperature, net immigration, and other factors during $^s\Delta_x(T(t))$[12,68]. Following the notation of Di Cola et al.[67] (page 523), the $i$th age class of stage $s$ is modeled as follows:

$$\frac{d\,^sN_i}{dt} = \frac{^sk \cdot \,^s\Delta_x}{^s\Delta}[^sN_{i-1}(t) - \,^sN_i(t)] - \,^s\mu_i(t)\,^sN_i(t). \qquad (1i)$$

In terms of flux, $^sn_i(t) = \,^sN_i(t)\,^s\nu_i(t)$ where $^s\nu_i(t) = \frac{^sk}{^s\Delta}\Delta_x(t)$, and

$$\frac{d}{dt}\left[\frac{^s\Delta\,^sn_i(t)}{^sk}\right] = \,^sn_{i-1}(t) - \,^sn_i(t) - \,^s\mu_i(t)\,^sn_i(t)\frac{^s\Delta}{^sk}. \qquad (1ii)$$

The total density in life stage s is $^sN(t) = \sum_{i=1}^{k}\,^sN_i(t)$.

Ignoring stage notation, new eggs enter the first age class of the egg-larval stage ($k = 1$), flow occurs between age classes and between stages, and surviving adults exit as deaths at maximum age ($i = \,^Ak$). The flow between age classes and stages is illustrated in Supplementary Fig. S11a. Absent mortality, the theoretical distribution of cohort developmental times (Supplementary Fig. S11b) can be estimated by Erlang parameter $k = \Delta^2/$var, where $var$ is the variance of $\Delta$. In our study, we used $k = 25$ or 40 for the egg-larval and pupal stages, while $k$ for the adult stage is equal the average longevity in days at 25 °C (Table 2 and Table S1). For example, if the average egg-larval stage is 128 $dd$, the standard deviation is 25.6 $dd$ for $k = 25$ and 20.2 $dd$ for $k = 40$. Furthermore, developmental times may vary with nutrition and other factors[31], and given appropriate data can be easily accommodated using the time varying form of the model[69]. Last, because of non-linearities and time varying nature, the model can only be evaluated numerically[12,70].

**Biodemographic functions (BDFs).** The general shapes of the BDFs are known (Fig. 1), hence the limiting values for temperature and relative humidity were extrapolations of the fitted functions to zero values[12,14].

*Developmental rates and times.* The time step in the population dynamics model is a day (d) of variable length in physiological time units. The development time in days ($d(T)$) varies with temperature ($T$), and the developmental rate function may be estimated by a simple nonlinear model ($R(T) = 1/d(T)$) (Eq. 2 and Fig. 1a). Ignoring the stage and time ($t$) variables,

$$R(T) = \frac{a(T - \theta_L)}{1 + b^{T-\theta_U}}, \qquad (2)$$

$a$ and $b$ are fitted constants, $\theta_L$ is the lower temperature threshold (i.e., $R(T) = 0$), and $\theta_U$ is the approximate upper inflection point, where the rate of development begins to depart strongly from linearity and rapidly declines to zero. Another similar model for $R(T)$ is also commonly used[71]. The developmental time constant ($\Delta$) in degree days (dd) was estimated in the linear range of favorable temperatures as $\Delta = d(T) \times (T - \theta_L)$, with the daily increment of physiological time at time $t$ computed as $\Delta_x(T(t)) = R(T(t))\Delta$.

Hence, a cohort initiated at time $t_0$ completes stage development on average when $\int_{t_0}^{t} R(T)dt = 1$ in continuous time, and $\sum_{t_0}^{t}\Delta_x(T) = \Delta$ in our discrete time model. In the field, $\Delta$ may change during the season as hosts of varying nutritional quality and ages become available[31], but lack of appropriate data precluded inclusion of this factor in the model.

*Reproduction.* Fruit fly females seek oviposition sites, and multiple attacks may occur per host. Realized oviposition ($S$) by all females is computed using the parasitoid form of a ratio-dependent demand-driven functional response model (Eq. 3)[12,72]. Absent a plant model, we assume a constant level of hosts ($H = 500$) that also keeps the model from increasing beyond reasonable bounds, and serves as a basis for comparisons across species and regions.

$$S = g(H, D, T)H = H\left(1 - e^{\frac{-D}{H}\left(1 - e^{\frac{-\alpha H}{D}}\right)}\right). \qquad (3)$$

$0 \leq \alpha = 0.005\Delta_x(T) < 1$ is the assumed proportion of $H$ discoverable during $\Delta_x(T)$[12,61,73], and $D$ is the time varying demand for oviposition sites by all adult females in the $k$ age cohorts($N(t) = \text{sr}\sum_{i=1}^{k}N_i(x, t)$) with assumed sex ratio sr $= 0.5$

(Eq. 4).

$$D = \phi_T(T) \cdot \phi_{RH}(RH) \cdot sr \sum_{x=1}^{k} f(x, T_{opt}) \cdot N_i(x, t), \qquad (4)$$

$f(x, T_{opt})$ is the oviposition profile of eggs female$^{-1}$ day$^{-1}$ at age ($i = x$, days) at optimum temperature ($T_{opt} = 25\,°C$) (Fig. 1b and Eq. 5) with fitted constants $\gamma$ and $\varphi$[74].

$$f(x, T_{opt}) = \frac{\gamma x}{\varphi^x} \qquad (5)$$

Concave BDFs ($0 \leq \phi_T(T(t)) \leq 1$, $0 \leq \phi_{RH}(RH(t)) \leq 1$) scale oviposition for the effects of mean temperature and mean % relative humidity respectively (Figures 1d,e), and may be viewed as survivorship functions (i.e., $\phi_T(T) \times \phi_{RH}(RH)$). N.B. Data for $\phi_T(T)$ and $\phi_{RH}(RH)$ were sparse or absent for some species, hence the lower and upper limits were deduced from the cited literature.

*Mortality*. BDFs for effects of temperature on daily mortality of stages are characteristically convex (Fig. 1c), and enters Eq. 1 as a component of the age ($i$) specific proportional mortality rate ($0<\mu_i(t)\leq 1$). Average daily temperature-dependent mortality rates were estimated from age specific life table data at different temperatures at ~50% survival, and from studies on the limiting effects of temperature extremes used to develop quarantine procedures. In some cases, polynomial functions were fit to better capture the data, with sufficient digits given to ensure accurate replication by others.

A measure of the favorability of temperature in each lattice cell is the average of the annual sums of daily mortality rates below the lower thermal threshold (i.e., $\bar{\mu}_{T<\theta_L} = (\sum_{i=1}^{y=years} \sum_{j=1}^{365d} \mu(T_{i,j} < \theta_L))/y)$, and similarly for $T$ above the upper threshold ($\theta_U$) (Eq. 2, Table 2, and Table S1).

**Summary of data and parameterization of BDFs**. Data to parameterize the BDFs (Fig. 1) were extracted primarily from age specific life table studies, but in some data were estimated from graphs and text summaries (Supplementary Excel files S1–S4).

**Mediterranean fruit fly**. Data to parameterize the BDFs for *C. capitata* are summarized in Fig. 5[43,51,75–82]. The lower thermal threshold for development is 10.29 °C for the egg stage, and 9.5 °C for the larval, pupal, and adult stages, and the upper thermal threshold for linear development is ~32.5 °C (Fig. 5a, b). The fly is moderately cold tolerant ($\mu(T)$, Fig. 5c, d). For example, the proportion of larvae surviving after 14 days at 2.2 °C is ~0.079, and ~0.0016 at 0 °C[83], with average daily survivorship rates of $0.834 = 10^{(\log 0.079)/14}$ and $0.631 = 10^{(\log 0.0016)/14}$, respectively. As mortality rates, the proportions dying d$^{-1}$ are $\mu(T = 2.2\,°C) = 0.166$ and $\mu(T = 0\,°C) = 0.369$. Per capita age-specific oviposition profiles at various temperatures[77] are illustrated in Fig. 5e, with lower and upper oviposition

thresholds being 15.5 and 32 °C, respectively ($\phi(T)$, Fig. 5f)[84]. Pupae are tolerant to low humidity[85], and adults exhibit higher desiccation resistance with lower rates of water loss under hot and dry conditions than other congeneric fruit flies, and have higher lipid reserves that are catabolized during water stress[3]. Reproduction and adult survival are high at RH > 33% (Fig. 1)[85], and are similar to that of olive fly and Mexican fruit fly. The scalar for the effects of relative humidity ($\phi(RH)$) on oviposition is shown in Fig. 5g. The effect of temperature on age specific fecundity is illustrated as 3-dimensional Fig. 5h.

**Melon fly**. The biology and host range of *B. cucurbitae* were reviewed[86], and data gleaned from the literature to parameterize the BDFs (Fig. 6 and Table 2) include: the thermal biology[78–80,87–92], the effects relative humidity[93,94], and low temperatures on fly mortality[95], and the effects of high temperatures on fecundity[96]. Some data were not used because of inconsistencies with other studies[92]. Melon fly has a relatively low thermal thresholds for larval development (7.95 °C) with a relatively high upper threshold of 33.5 °C (Fig. 6a, b), making it moderately tolerant to both low and high temperatures[88]. The fly has relatively low fecundity (Fig. 6c) with a skewed bias to higher temperatures (($\phi(T)$, Fig. 6d)[88]. Data were not available for the effects of RH on reproduction, but high relative humidity is favorable for pupal development and adult reproduction[93,94] ($\phi(RH)$, Fig. 6e). These attributes conform to the observations that melon fly is reproductively most active during the cooler, yet hot periods with high humidity that occur during the monsoon season of mid to late summer in south Asia[94]. The egg stage has a narrower tolerance to temperature ($^E\mu(T)$, Fig. 6f) than the larva-pupal stages ($^{L-P}\mu(T)$, Fig. 6g)[95,96]. Figure 6h is a 3-dimensional representation of age-specific fecundity as affected by temperature.

**Oriental fruit fly**. The data to parameterize *B. dorsalis* developmental rates and fecundity as affected by temperature and humidity[75,78–80,97,98], and the thermal mortality rates[46,95,99–101] are illustrated in Fig. 7. The lower and upper thermal thresholds for the life stages vary 8.87-10.2 °C and 35–36 °C respectively (Fig. 7a–c), the fly has a long preoviposition period (Fig. 7d), and an oviposition bias toward the lower part of its 15.5–36 °C range (($\phi(T)$, Fig. 7e)[89]. We did not use data from an inbred laboratory colony reared on artificial diet for 200 generations[102] that predicted a more symmetrical $\phi(T)$ with narrower thermal limits of 16.7–34.9 °C. Experimental data were not available for the effects of RH on reproduction, and hence trap catches (a measure of adult activity) that were positively correlated with maximum RH but negatively correlated to minimum RH[103] were used to infer the symmetrical concave $\phi(RH)$ with limits 40–95% (Fig. 7f). The fly is relatively cold tolerant ($\mu(T)$, Fig. 7h)[46,95,99–101]. Figure 7g is a 3-dimensional representation of age-specific fecundity as affected by temperature.

**Mexican fruit fly**. Data from age specific life tables[104–108] were used to estimate developmental rates, and the fecundity profile for *A. ludens*. Data on the

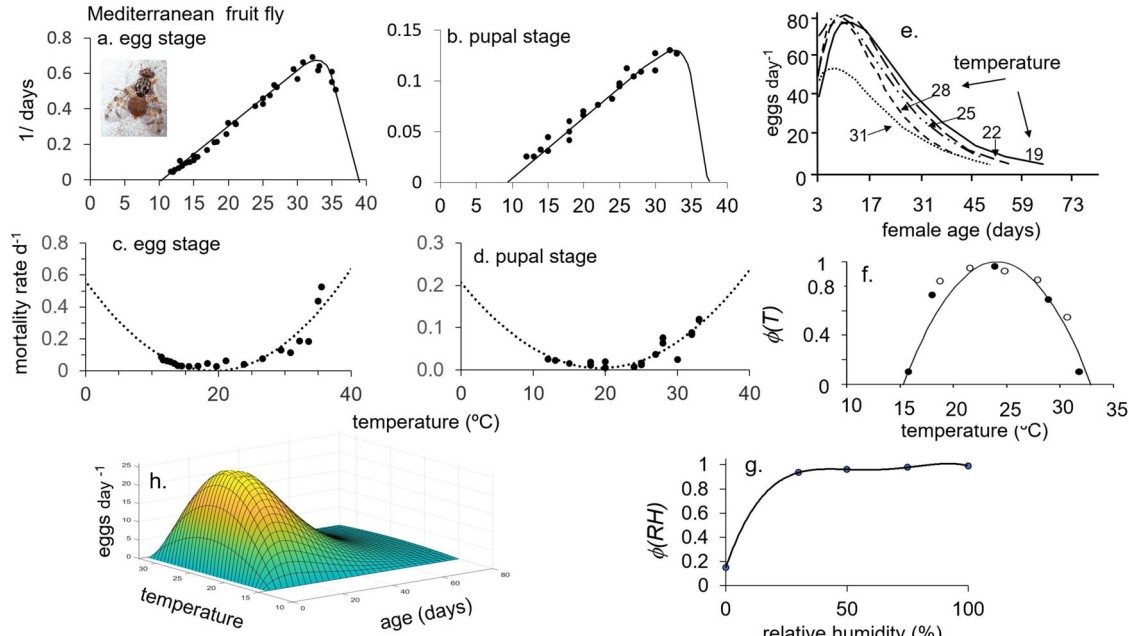

**Fig. 5 Biodemographic functions for Mediterranean fruit fly. a, b** Rates of development of the egg[75,81] and pupal stages respectively, **c, d** mortality rates for egg and pupal stages respectively[43,51,75,76,78,79,81,82], **e** per capita fecundity profiles on age in days at different temperatures[77,83], **f, g** temperature ($\phi(T)$)[84] and relative humidity ($\phi(RH)$)[85] scalars for adult reproduction, and **h** 3-D age specific fecundity profiles corrected for the effects of $\phi(T)$. The clip of Mediterranean fruit fly is by Jack Kelly Clark, University of California Statewide IPM Program.

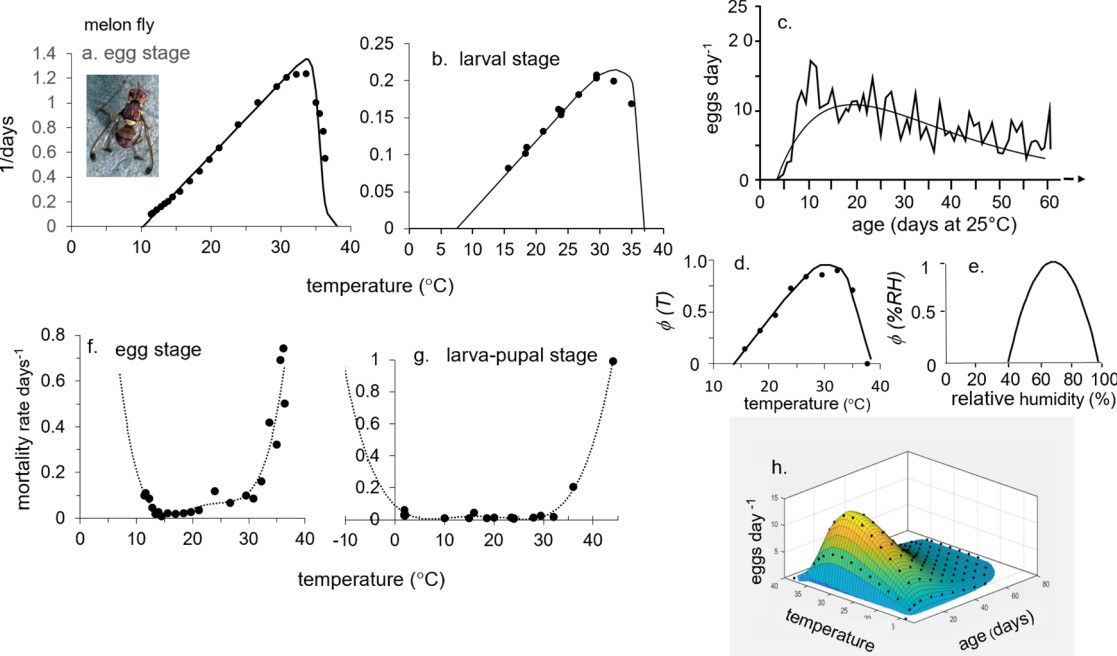

**Fig. 6 Biodemographic functions for melon fruit fly. a, b** Rates of development of the egg and larval stages[78–80,87–92], **c** per capita fecundity profile[88] on age in days at 25 °C, **d, e** temperature ($\phi(T)$)[88] and inferred relative humidity ($\phi(RH)$)[93,94] scalars for reproduction, **f, g** mortality rates for egg stage and for larval-adult stages respectively[95,96], and **h** 3-D age specific fecundity profiles corrected for the effects of $\phi(T)$. The clip of melon fly is by Jack Kelly Clark, University of California Statewide IPM Program.

developmental rate of the combined egg-larval-pupal stages on temperature are shown in Fig. 8a. The lower and upper thermal thresholds for development are 10.3 °C and 34 °C, respectively. Two similar shaped age specific fecundity profiles were published (Fig. 8b)[107,108], but we used the more conservative one[107]. The preoviposition period is 10–15 days at ~25 °C, and oviposition is assumed to occur in the range 12.5–32.5 °C with a peak at 22.65 °C (($\phi(T)$, Fig. 8c). Figure 8e is a 3-dimensional representation of age-specific fecundity as affected by temperature. Mexfly is 36% more desiccation resistant than medfly[3], and suffers little loss of reproductive capability at low humidity[109], hence the BDF ($\phi(RH)$, Fig. 8d) was scaled up from medfly values (Fig. 5g). Data on mortality rates ($\mu(T)$) at low temperatures were estimated from early bioclimatic studies[109,110]. Survival of mexfly at 0.97 and 44 °C were compared to that of medfly and oriental fruit fly[105], with the order of tolerance at 0.97 °C being *B. dorsalis* > *C. capitata* > *A. ludens*, and tolerance to 44 °C is *B. dorsalis* > *A. ludens* > *C. capitata*. The fly is largely limited to frost-free areas[111], with high mortality d⁻¹ assumed high at 0 °C (Fig. 8f).

**Weather data**. Available historical climate weather data for North and Central America and the European-Mediterranean region for years 1980–2010 include daily maximum and minimum temperature, rainfall, relative humidity, and solar radiation (W m⁻² day⁻¹), but only temperature and RH are used in the models. The weather data are from AgMERRA[112], a global baseline forcing dataset of the Agricultural Model Inter-comparison and Improvement Project (AgMIP, http://www.agmip.org/) that is a reanalysis of weather observations[113] combined with observational datasets from in situ observation networks and satellites[112]. The AgMERRA weather data (https://data.giss.nasa.gov/ impacts/agmipcf/) have a ~25 km spatial resolution for each of the 15,843 lattice cells for the USA, Mexico, and Central America, and 17,791 lattice cells for the Euro-Mediterranean region.

Climate change weather data for the European-Mediterranean region was downscaled from coarser ~200 km⁻² resolution global climate model data[114] to a ~30 km resolution using the PROTHEUS regional climate model[115]. PROTHEUS is a coupled atmosphere-ocean regional model that allows simulation of local extremes of weather via the inclusion of a fine-scale representation of topography and the influence of the Mediterranean Sea[115]. This enables increasing the spatial resolution and accommodating poikilotherm sensitivity to fine-scale climate and local topography effects. Specifically, we used daily max–min temperatures and RH from the A1B regional climate change scenario that is towards the middle of the IPCC[22] range of greenhouse gas (GHG) forcing scenarios[116]. The uncertainty associated with climate model predictions forced using the A1B scenario is low for the Mediterranean region relative to the rest of the globe[117]. A subset of daily weather for the period 1 January 2040 to 31 December 2050 for 10,598 lattice cells was used for the Euro-Mediterranean region.

Daily climate change simulations for North and Central America weather are from the NASA Earth Exchange Global Daily Downscaled Projections (NEX-GDDP) dataset[118] at ~25 km resolution (https://www.nccs.nasa.gov/services/data-collections/land-based-products/nex-gddp), derived from global climate simulations of the Max Planck Institute Earth System Model low resolution (MPI-ESM-LR) model[119] using a statistical downscaling technique. Global MPI-ESM-LR climate simulations were forced by the Representative Concentration Pathway 8.5 (RCP 8.5) scenario[120] that corresponds to a range of warming similar to A1B[121]. An evaluation of historical simulations of North American climate using continental metrics of bias relative to observed weather[120], showed that MPI-ESM-LR is the top ranked among the core set of 17 global climate models considered. Based on this downscaled climate change scenario, a daily weather dataset was developed for the period 1 January 2045 to 31 December 2075 for 20,355 lattice cells in North and Central America.

Daily relative humidity data for climate change weather scenarios were not available for North and Central America and were computed as $\%RH = 100 \times e_m/e_s$, where $e_s$ is the saturated vapor pressure and $e_m$ is ambient vapor pressure[121]. Specifically, $e_s = 610.78 \exp(17.269 T_{mean}/(237.3 + T_{mean}))$, and $e_m = 610.78 \exp(17.269 T_{min}/(237.3 + T_{min}))$ assuming $T_{min}$ approximates the dew-point temperature.

**GIS mapping and marginal analysis**. The PBDMs were run continuously for each lattice cells using daily weather data for the specified periods. Although daily age structure dynamics of the flies are predicted by the model for each lattice cell, only yearly georeferenced summary data for all variables for each lattice cell were written to year-specific text files, and at the end of multiyear runs, means, standard deviations, and coefficients of variation were computed omitting the first year when the model was assumed equilibrating. Average cumulative pupae per year below 2000 m of elevation were used as a metric of favorability for each lattice cell. No calibrations of the models were made to fit common-wisdom notions of the geographic distribution and abundance of a species.

The open-source geographic information system (GIS) GRASS originally developed by the United States Army Corps of Engineers was used for geospatial data management, analysis and mapping using bicubic spline interpolation on a 3 km raster grid to match the resolution of the underlying digital elevation model. The interpolated PBDM raster maps were overlaid on base map layers made using GRASS GIS with Natural Earth free vector (administrative boundaries, the Great Lakes) and raster (shaded relief) map data available in the public domain (https://www.naturalearthdata.com/). The digital elevation model used in GIS computations (e.g., for restricting mapping of PBDM raster maps below a specified elevation) is the public domain NOAA Global Land One-km Base Elevation (GLOBE; https://www.ngdc.noaa.gov/mgg/topo/globe.html). GRASS is maintained and further developed by the GRASS Development Team (see http://grass.osgeo.org)[122]. GRASS version 7.9 was used in the study[16].

Often greater insights can be gained using econometric techniques. For some species, multiple linear regression (Eq. 6) with unknown error term $U$ was used to summarize the simulation data, and to explore the impact of weather and

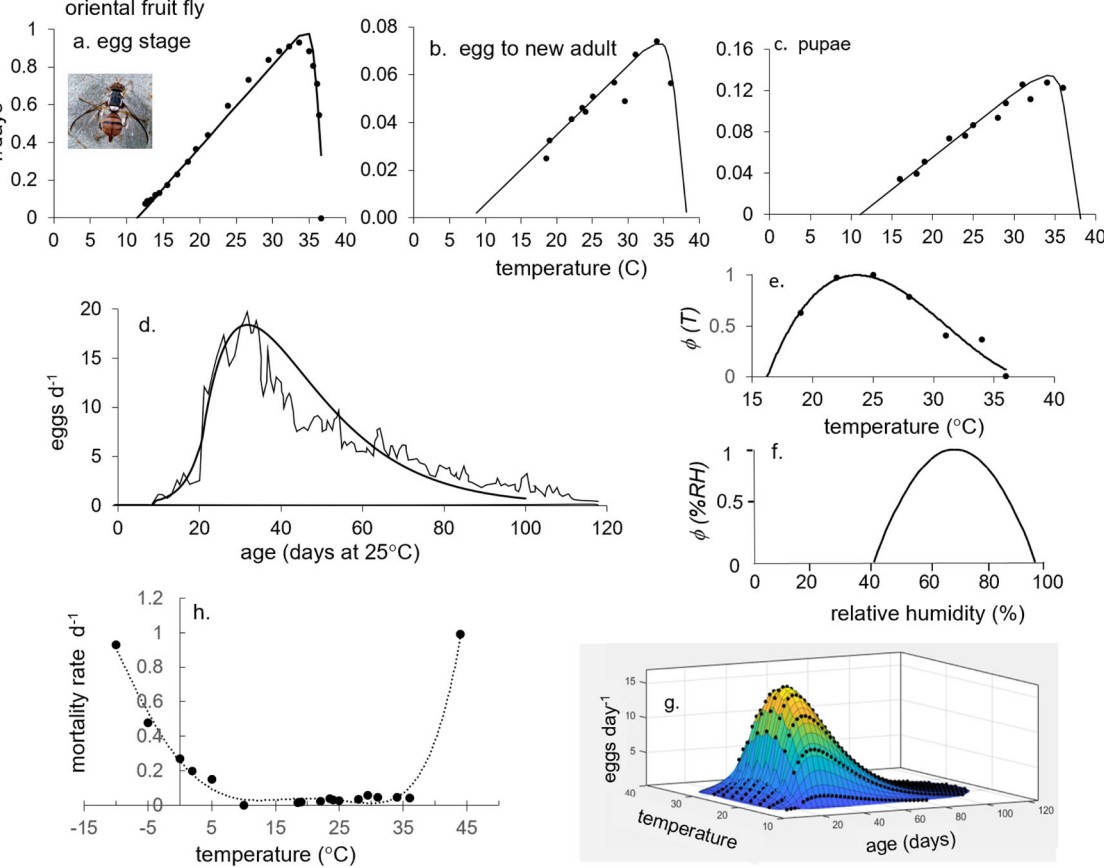

**Fig. 7 Biodemographic functions for oriental fruit fly. a–c** Rates of development of the egg, egg to new adult, and pupal stages respectively[75,78-80,97,98], **d** per capita fecundity profile on age in days at 25 °C[98], and the inferred scalars for reproduction on **e** temperature ($\phi(T)$)[79,80,89] and **f** relative humidity ($\phi(RH)$)[103], **g** 3-D age specific fecundity profiles corrected for the effects of $\phi(T)$, and **h** mortality rates for all life stages[46,95,99-101]. The clip of oriental fruit fly is by Jack Kelly Clark, University of California Statewide IPM Program.

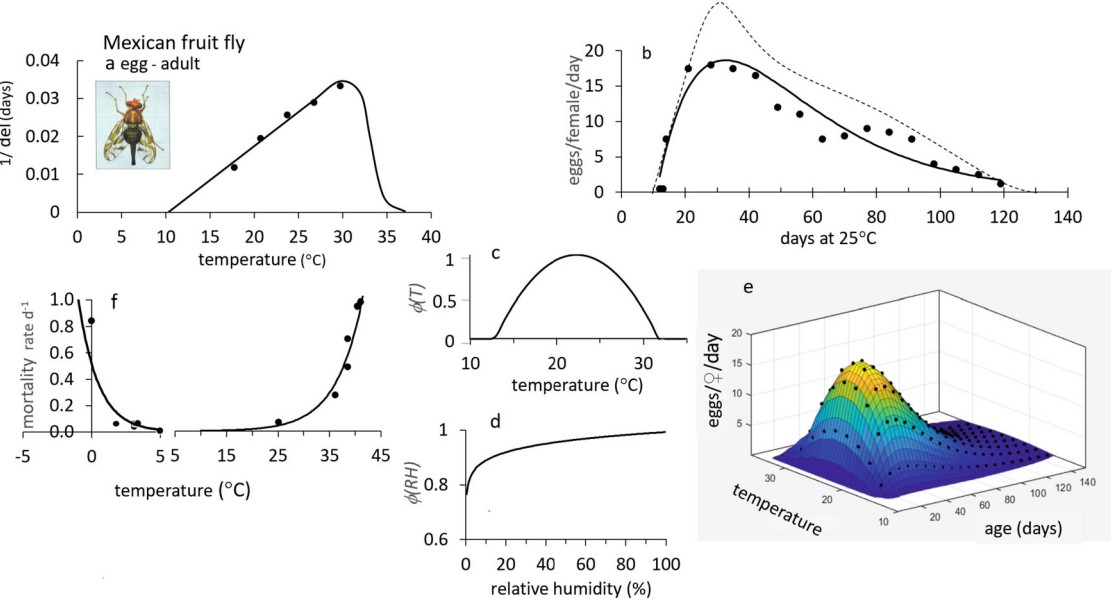

**Fig. 8 Biodemographic functions for Mexican fruit fly. a** Rates of development of egg-pupal stages[104-108], **b** per capita fecundity profile on age in days at 25 °C (———)[107] and (------)[108], **c, d** temperature ($\phi(T)$) and relative humidity ($\phi(RH)$)) scalars for adult reproduction with $\phi(RH)$ scaled up 36% from that reported for medfly, **e** 3-D age specific fecundity profiles corrected for the effects of $\phi(T)$, and **f** mortality rates for all life stages. The clip of Mexican fruit fly is by Jack Kelly Clark, University of California Statewide IPM Program.

computed variables ($x_i$) and the $J$ interactions ($x_j$) on the annual total number of pupae produced.

$$\text{pupae} = f(x_1, \ldots, x_n) = a + \sum_i^I b_i x_i + \sum_j^J b_j x_j + U \tag{6}$$

To estimate the marginal effects of specific factors ($x_i$), the partial derivative of Eq. 6 with respects $x_i$ is computed, using the averages of remaining independent variables $\left( \frac{\partial \text{pupae}}{\partial x_i}, i = 1, \ldots, I \right)$.

**Reporting summary**. Further information on research design is available in the Nature Research Reporting Summary linked to this article.

## Data availability

All biological data used in the analysis are publicly available or were sourced from tables, text or estimated from figures in the cited literature. All data from the diverse sources are shown in the fitted biodemographic functions plotted in Figs. 5–8. The data used to parameterize the BDFs are available as Supplementary Excel files S1–S4. Base geodata layers used to generate maps are available in the public domain as Natural Earth vector and raster data (https://www.naturalearthdata.com/) and GLOBE digital elevation model data (NOAA Global Land One-km Base Elevation, https://www.ngdc.noaa.gov/mgg/topo/globe.html).

## Code availability

The algorithms (code) used in the present PBDM analysis is available from the corresponding authors on reasonable request. The source code is in Borland Pascal embedded in a larger code base of about ten thousand lines that includes PBDMs for >60 different species of plants, herbivores, parasitoids, predators, and pathogens, a subset of which have been published as PBDM analyses implemented in a GIS context[123]. The code for fruit flies is currently being rewritten in Python using the object-oriented programming paradigm for release as open source[124]. The Pascal PBDM code developed over the last three decades is currently not licensed nor deposited in a code repository. The code is managed by the non-profit scientific consortium Center for the Analysis of Sustainable Agricultural Systems Global (CASAS Global, http://www.casasglobal.org/). The Pascal subroutine for the distributed maturation time dynamics model with and without attrition, has been published[12] (pages 157–159), and all biodemographic functions are available in Table 2 and Supplementary Table S1.

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

## Acknowledgements

We thank Dr. N.C. Manoukis (USDA) and an anonymous reviewer for constructive suggestions that improved the manuscript. Dr. Donald Thomas (USDA) was an invaluable source of information on Mexican fruit fly. We continue to be grateful to the international network of developers who maintain and continue to improve the Geographic Resources Analysis Support System (GRASS, https://www.grass.osgeo.org) GIS software, and make it available to the scientific community. We thank Bridget Thrasher from the Climate Analytics Group (http://www.climateanalyticsgroup.org) for help in obtaining the NASA climate model weather data. The climate scenario for the USA and Mexico are from the NEX-GDDP dataset, prepared by the Climate Analytics Group and NASA Ames Research Center using the NASA Earth Exchange, and distributed by the NASA Center for Climate Simulation (NCCS). The climate scenario for the Euro-Mediterranean region was developed and provided by the Laboratorio Modellistica Climatica e Impatti at ENEA, Rome, Italy. The study was supported by the Center for the Analysis of Sustainable Agricultural Systems Global (CASAS Global, http://www.casasglobal.org/), by Agenzia nazionale per le nuove tecnologie, l'energia e lo sviluppo economico sostenibile (ENEA), Rome, Italy, and by the project MED-GOLD funded by the European Union's Horizon 2020 research and innovation programme under grant agreement No 776467. Clips of the tropical fruit flies in the figures are from photographs by Jack Kelly Clark provided courtesy of the University of California Statewide IPM Program.

## Author contributions

A.P.G and L.P. conceived and designed the work, performed the PBDM/GIS analysis, and with D.M.S. wrote the initial draft of the manuscript. L.P. developed the weather data to run the PBDMs, and M.N. provided expert help with GIS issues. J.R.C. did extensive rewriting of PBDM Pascal computer code into Python. All authors contributed to the interpretation of data, substantively revised the manuscript, and have approved the submitted version.

## Competing interests

The authors declare no competing interests.
