## [Peer Review File · Communications Biology]

Reviewers' comments:

Reviewer #1 (Remarks to the Author):

The paper by Gutierrez et al, "Invasive potential of tropical fruit flies in temperate regions", addresses some of the most serious invasive pests threatening agriculture world wide- tephritid fruit flies. Specifically, the authors aim to model the potential geographic range of key species via PBDMs ("Physiologically-Based Demographic Models") as well as potential shifts to these ranges under possible climate change scenarios in order to inform policy and action program planning. Overall the paper is very well written and the analysis highly interesting, as I would expect from these authors- some of whom are among the best in their fields, and with long experience. However, there are some improvements and details to be added that I think would benefit the work before acceptance.

In terms of the context for this study as laid out in the abstract, introduction, and discussion, the authors repeatedly state that programs do not sufficiently consider weather when triggering quarantine or eradication programs. Example: "Ongoing concern about the potential of tropical fruit flies to invade agriculture has led to a substantial development of surveillance and eradication infrastructure, mostly without thorough understanding of weather-related constraints to invasion" (lines 54 – 57), and similar. We can quibble about the meaning of the word "thorough" above, but I would argue that overall the statement is false. Though I am a researcher and not a program manager, I have some familiarity with the fruit fly detection and eradication procedures in California, and I know that weather plays a critical role in temporal and spatial trapping patterns, quarantine length (via 3 generations of degree-day calculation), and application of SIT- and probably more.

From a research perspective, the notion that the impact of weather (and climate) on Tephritid fruit fly invasions and establishment has not been widely investigated is also untrue. These have been the subject of study for a century in fruit flies; prominent early examples include the research of Messenger, Flitters and colleagues in the mid-1950's, and this vein has continued since right up to the work of two of the authors (Biol. Invasions 13 (12) 2661-2676, 2011). In recent years this research has included "standard" approaches such as MAXENT and CLIMEX as well as more esoteric efforts with custom agent-based simulations (ABS) (full disclosure, ABS mentioned is my own work). The authors are clearly aware of this as they cite these studies.

Tropical fruit flies are among the best studied invasive pests in the world: A search in Dimensions (<https://www.dimensions.ai/>) using the terms "Mediterranean fruit fly" and "invasion" and "weather" yields 4303 publications since 1957 (note- this is just one of many tropical fruit flies of concern). The same search with some of the most notorious invasive pests in the world shows how well studied tephritidae are: "Asian long horned beetle" gets 118 publications, "brown marmorated stink bug" returns 162, and "Kudzu" has 2284. The only example I found that was similar was "Japanese Beetle", with 4541, perhaps because it has, similar to medfly, been studied for many years.

Having written the (admittedly long-winded) defense of the research and program applications relating to the impact of weather and climate on fruit fly range above, I stress that the contribution from this paper is no less useful or important. I think application of PBDMs is relatively infrequent for fruit flies (only 37 publications in Dimensions, and most include some of the present authors!) and is helpful. Proper and accurate context is essential, however, for the reader to be able to understand where these fit in and also to avoid distortion of the scientific record.

Concern about the realism of the favorability indices that are shown in various figures: I don't think I saw any attempt to "ground truth" these. Is there a way to estimate if these are realistic? This is fought, but perhaps based on detection data? I think the authors should make some attempt to do this, or the application of these modeled results will be limited. This theme repeats below.

Regarding the analysis, it is cutting-edge and performed by experts. The parametrization work alone for four species is difficult, and what they have accomplished is impressive. How might this

be replicated by another researcher? I did not see where the authors deposited their models and parameter sets so that they can be explored by others- I may have missed this, but if it is not been done via a metadata server or similar, it should be. Software versions should also be included for replicability purposes. In general repeatability/verifiability was a concern for me: Even after reading the paper and SI I would have no idea how to recreate what is presented. I encourage the authors to be specific: What software was used, how can I obtain it to run these analyses? Again, I might have missed this.

Perspective distributions of fruit flies are given in various figures for the period 1980 – 1990 and then for 2040 – 2050 and later under climate change scenario. Weather data mentioned in M&M (pg 19) indicates historical records for 1990 – 2010 and modeled data for 2040 – 2050 and 2045 – 2075 (pg 20). This is confusing, I suggest authors clarify sources and date ranges of data.

Discussion opens with a critique of correlative methods that include occurrence data. There is also this: “we need to move beyond repetition of loose verbal arguments of invasive potential that ignore objective criteria or that are based on statistical inference of detection data that ignore weather effects” (lines 258 – 260). I agree! Can observational data be used for the species studies to falsify the distribution of suitabilities that are presented for these four species? This is an echo of my comment on realism above.

Finally, a question that kept recurring as I read the manuscript: How can the information presented benefit the programs in question? What would the authors suggest should be changed as a result of this analysis? From practical experience here in Hawaii it is difficult for me to imagine that agencies over-reacted to incursion of these species in portions of S. California, where I lived for a decade not too long ago- conditions on the Kona side of Hawaii island are quite similar, and medfly, oriental fruit fly, and melon fly (plus, recently, olive fly!) are all easy to find there. Ground truthing (“validation”) of the sort I suggest above would do a lot to help move this argument forward.

All my comments are made in the spirit of improving the work, and I hope they are received in the same way. I would be happy to review a revision. Should this paper be accepted I will gladly accept an acknowledgement of my signed review.

NC Manoukis
USDA-ARS
Hilo, Hawaii USA

Smaller comments

Lines 52-54: While many will agree, this statement is too sweeping; particularly without any citation to a supporting study.

Table 1, should “host specificity” of Olive fly indicate “monophagous” to qualitatively match the other entries? It is also not clear to me what criteria were used to establish “sufficiency” of the data for the table- this should be explained.

Reviewer #2 (Remarks to the Author):

“Invasive potential of tropical fruit flies in temperate regions” is a complete and meticulous paper about the use of PBDMs to predict the spreading of invasive tropical fruit flies in North and Central Americas and Europe, underlining the threat these fruit flies represent, favoured by climate change especially.

I have some comments and suggestions here reported.

Some integration about the 4 fruit flies spreading and record in Europe and Americas are essential; information in supplementary materials is not complete too (what about di current distribution of medfly in Europe or MB?). The records of tropical fruit flies outside their native (or already invaded) countries represent crucial information. Authors should add some papers about the

finding in fields, highlighting the simple records mentioned in EUPHRESCO and EFSA reports.
Line 64: please take into account recent interceptions of *B. dorsalis* in Europe (Nugnes et al., 2018; Egartner and Lethmayer, 2017)
Lines 74-76: please report the descriptor of the mentioned species.
Lines 77-78: reference needed

Figure 1: please change "fruit flies" in "flies" or "tephritids and Drosophilidae". *Drosophila* is commonly defined a fruit fly but belongs to a totally different family.

Table 1. about the host specificity: stenophagous is in the table legend although no "steno" was mentioned. I would suggest avoiding the explanation (Polyphagous and Specific host preference are defined already).

In my opinion methods, results and supplementary material should be reassembled. Supplementary material reported a large amount of information that needs to be moved in Methods or result. For example, I would synthesize the biological parameters of each studied fruit fly in a table, reporting the relative references too. In this way Supplementary material could be shorter and information to which authors referred to could be moved in the main text. Same arrangement could be useful in results set-up also.

Figures 3-4-5-s2-s4-s6-s8: light blue could lead to misunderstanding because it is not clear enough if the light blue moves from white (0 value) or is near to the yellow one (mid-high value). I hope authors can change the colours pattern.

In supplementary material,

- Legend years did not correspond to years in the perspective distribution (example as in fig s11: NA-CA 2050-2060 in the legend and 2055-2065 in the figure). Authors should check the whole manuscript also to avoid this error.

- In all perspective distributions, authors made some comparison between 1980-1990 and 2050-2060 distribution but, as reported in the legends, the first set of picture referred to pupae the second to the fly. Is this the right meaning of the comparisons?

Discussion should be integrated with some literature in order to compare the obtained results with recently published ones. Please see the list I suggested.

De Villiers, M.; Hattingh, V.; Kriticos, D.J.; Brunel, S.; Vayssières, J.F.; Sinzogan, A.; Billah, M.K.; Mohamed, S.A.; Mwatawala, M.; Abdelgader, H.; et al. The potential distribution of *Bactrocera dorsalis*: Considering phenology and irrigation patterns. *Bull. Entomol. Res.* 2015, 106, 19–33.

Egartner, A.; Lethmayer, C. Invasive fruit flies of economic importance in Austria—Monitoring activities. *Integr. Prot. Fruit Crops IOBC-WPRS Bull.* 2017, 123, 45–49.

Nugnes, F., Russo, E., Viggiani, G., & Bernardo, U. (2018). First record of an invasive fruit fly belonging to *bactrocera dorsalis* complex (Diptera: Tephritidae) in Europe. *Insects*, 9(4).

<https://doi.org/10.3390/insects9040182>

Pieterse W, Terblanche JS, Addison P. Do thermal tolerances and rapid thermal responses contribute to the invasion potential of *Bactrocera Dorsalis* (Diptera: Tephritidae)? *J Insect Physiol.* 2017;98:1–6.

Qin, Y., Wang, C., Zhao, Z., Pan, X., & Li, Z. (2019). Climate change impacts on the global potential geographical distribution of the agricultural invasive pest, *Bactrocera dorsalis* (Hendel) (Diptera: Tephritidae). *Climatic Change*, 155(2), 145–156. <https://doi.org/10.1007/s10584-019-02460-3>

Minor changes:

Pro capita should be written in italic, please check the whole manuscript

Line 240: biotic?

Line 278: "in our study"

Supplementary information: "dd" is not always reported, please check

Reviewers' comments of Decision on manuscript COMMSBIO-21-0038-T

Invasive potential of tropical fruit flies in temperate regions

Andrew Paul Gutierrez, Luigi Ponti, Markus Neteler, David Maxwell Suckling, Jose Ricardo Cure

Reviewer #1 (remark to Dr. Manoukis):

We appreciate very much the signed comments of fruit fly expert Dr. Manoukis (whom we do not know except from the literature), and the spirit in which they were given. The focus of our paper was to show how the weather driven biology of the four fruit flies as captured by parameterized biodemographic function estimated from data in the literature could be used to predict their geographic distribution and relative abundance. The PBDM approach differs from standard correlative methods that rely on occurrence/detection data that may be of questionable value with regards establishment to find correlates in aggregate weather data to estimate the species climatic envelope. As we state, PBDM are weather driven physiologically based time varying life tables. Further, we note that other species of fruit flies could be incorporated using the same template models.

The word ‘thorough’ in the text was highlighted by Dr. Manoukis is appropriate because ‘... [a] thorough understanding of weather-related constraints to invasion’ has not been part of the modus operandi of eradication agencies. I (APG) have been in contact with USDA and California Department of Agriculture staff and discussed their methods – the real question is how they can use the weather data in their projections. However, we can use softer language. Further, P.S. Messenger, whom Dr. Manoukis cites, was an early proponent of studying the effect of weather on the dynamics and distribution of pest species – he was one of my (APG’s) early mentors. Also, the first PBDMs were developed by Gilbert, N. E. and A. P. Gutierrez. (1973. **A plant-aphid-parasite relationship. J. Anim. Ecol. (42): 323-340**). We are well aware of the use of correlative methods such as Maxent and CLIMEX to predict the potential geographic range of species, but the use of these methods has been criticized as they depend on species occurrence data that are often of questionable reliability (see text) – that often do not tell us if they were detection data or of establishment of the species. In fact, the development of CLIMEX was based on early work by H.A. Nix and APG (e.g., **Gutierrez, A. P., D. E. Havenstein, H. A. Nix and P. A. Moore. 1974. The ecology of *Aphis craccivora* Koch and subterranean clover stunt virus. III. A regional perspective of the phenology and migration of the cowpea aphid. J. Appl. Ecol. (11): 21-35.**) (e.g., the citations in R. Suthurst’s original paper reporting CLIMEX). I did not pursue this approach because it informs nothing about the dynamics. Of the two correlative approaches, MaxEnt is more robust than CLIMEX, and but both are useful approaches. However, if the detection records from California for the four species were used in a correlative analysis (e.g., CLIMEX), parts of California would be included in the climatic envelope, and yet the person in charge of the eradication efforts in California (Dr. Kyle Beucke, CDFA, Primary Entomologist, pers. comm.) informed us via email that none of the four fruit flies in our study are established in California, this despite multitude detections (see text).). However, the model provides insights about the increased likelihood of establishment of the four species in California (and elsewhere) as California becomes more tropical with climate change.

I am less aware of Dr. Manoukis’ custom agent-based simulations (ABS) approach (individual-based models (IBMs)) that I think would be useful in behavioral studies, but we find it difficult to image how it

could be scaled up to a regional level – I will look into the matter further. I don't think ABS applies to our paper, but it can be cited as an alternative approach.

We did a very comprehensive search of the literature that we summarized in the text as “*consisting of a panoply of studies on age-specific life table studies, on field ecology, reproductive maturation and host preferences, thermal responses, mating behavior, temperature treatments designed for quarantine measures, and academic topics such as aging in fruit flies – of incomplete data sets gathered for disparate purposes. The issue then is how to come to grips with this cacophony of partial information.*” It was a very difficult task to sort the data and examine them for inconsistencies. To quote Dr. Manoukis, “The parametrization work alone for four species is difficult, and what they have accomplished is impressive.”

An important question Dr. Manoukis raises is: **“What software was used, how can I obtain it to run these analyses?”** The models are parameter sparse and the mathematics are presented in full in the text (eqns. 1-6) and all of the biodemographic functions and parameter estimates are presented in the tables 2 and 3 in the supplementary materials. No special software was used or required – it was programmed in PASCAL as a population dynamics model in a input-output system for models of more than 40 species that are run using daily (AgMERRA) weather data that in our study was at ~25 km² spatial resolution for each of the 15,843 lattice cells for the USA, Mexico, and Central America, and 17,791 for the Euro-Mediterranean region. The PASCAL code for doing the Euler integration is found in Gutierrez 1996 p. 157 as cited. The massive weather data is available from the cited sources. The open-source GIS software GRASS was used to map the simulation results (NB one of the authors M. Neteler is a key developer of the open-source GRASS system). The only question that might arise is how the daily mortality rates on temperature and RH were estimated. Specifically, they are point estimates in the linear range of survivorship data reported in age specific studies or estimated from studies on thermal death at low and high temperatures. All of the simple statistics were computed in EXCEL or with standard statistical software. Having explained this, we have long recognized that it would be useful to develop software that others, without skills in population dynamics and programming, could use in such analyses, but to automate these calculations is a very large task indeed, and funding to generalize the software has not been available. **However, we are in the process of developing open-source python-based object-oriented software to do this that we will make available.** Leadership of python code development is by the new author J.R. Cure. All of the research we report in this paper was largely unfunded – it was a collaborative effort by CASAS Global associates.

The question about the time frames of weather records is easily resolved as the time frames of weather available to us and that actually used in the simulations. Dr. Manoukis asks if the observational data can be used to falsify the distribution of suitability that are presented for these four species? Until those data are summarized in a georeferenced manner and the veracity of the record determined, such comparisons are of dubious value. We qualitatively made comparisons for medfly, Oriental fruit fly and Mexican fruit fly by comparing maps by other authors who used correlative method to our PBDM projections (see Supplemental materials). Those maps tend to be similar but geographically wider than our results. We can however, do site and time specific analyses that correlative methods cannot (see references in the paper for *Tuta absoluta* and other species).

Dr. Manoukis' comment that **“From practical experience here in Hawaii it is difficult for me to imagine that agencies over-reacted to incursion of these species in portions of S. California, where I lived for a decade not too long ago- conditions on the Kona side of Hawaii island are quite similar, and medfly, oriental fruit fly, and melon fly (plus, recently, olive fly!) are all easy to find there. Ground truthing (“validation”) of the sort I suggest above would do a lot to help move this**

argument forward.” This is exactly, this kind of intuitive extrapolations that we sought to dispel, as the ectothermic response to weather is different than our perceptions of suitability. Specifically, the model examines the favorability of weather from the fly’s point of view and not our qualitative assessment of favorability, and that includes short term weather patterns that may be limiting. The model agrees with Dr. Manoukis’ observation of the establishment of three of the species in Hawaii with good potential for the establishment of Mexican fruit fly there, but the model predicts the four species have low likelihood of establishment in California under extant weather. Finer grain weather data would be required to map the distribution in Hawaii. Prior versions of the model predict the establishment of olive fly in Hawaii and California. Dr. Manoukis’ comment provides the ground truthing for Hawaii. The arguments of Papadopoulos et al. (2013) that several species of tropical fruit flies are established at undetectable levels in California is verification of model that predictions of low favorability. We cannot falsify Papadopoulos et al. supposition that has a questionable basis. These observations fall under the verbal ambit of the unverifiable notion of "fruit fly friendly area". To this end, Oriental fruit fly is often trapped in southern California, and we developed a special analysis in the Supplemental Information using the capture and simulation data to explain its lack of establishment (SI Fig. S3-S5

Finally, Dr. Manoukis asks an important question – **“How can the information presented benefit the programs in question? What would the authors suggest should be changed as a result of this analysis?”**

About five years ago, APG gave a two-day conference to the National USDA invasive species group in North Carolina and via video to USDA nationally. The response was support from the scientific staff present in NC, but silence from the group leaders who appeared to understand little of what had been discussed. So, until such groups are headed by forward looking scientist, little can be done other than to challenge the current system with new analyses. Getting researchers like Dr. Manoukis involved in collaborative projects would resolve many of the questions about the utility of the PDBM approach – we are very open to such fruitful collaboration.

Reviewer #2 (Remarks to the Author):

In some respects, the issues raised by reviewer # 2 are more difficult despite also being complementary.

The reference to new records of the fly have been integrated in the text, but they tell us little about their establishment.

The color scheme is one developed after much effort, and the reviewer is the first to raise this issue – the patterns are clear.

The request to reassemble the paper putting more of the details given in the supplementary material into the methods has been accomplished leading to a delay in revising the paper. This change improved the paper and makes it more readable. These changes do not change the substance of the text. The paper has been cast to conform to journal requirements.

The years for the weather data have been standardized – corrected.

The suggested references have been integrated into the text.

The other comments are of technical nature and were easily considered in a revision.

For the authors,

Andrew Paul Gutierrez FRES

REVIEWERS' COMMENTS:

Reviewer #1 (Remarks to the Author):

Dear Authors-

I appreciate your thorough (there is that word again! But here I think it is indisputable) and detailed rebuttal. My thanks also for the careful and complete explanation of where the PBDM approach sits relative to other approaches such as CLIMEX, Maxent. I think the paper is much improved.

In reading this version, I suggest acceptance - the paper is clear, complete, and well presented. The results should be useful to entities like CDFA in focusing their surveillance efforts and calibrating their response to incursions/ detections.

Code availability section is fine, neatly handles my question- thank you. I am looking forward to learning more about the Python implementation described, as it becomes available.

The question of ground truthing is thorny, and I am certainly familiar with the arguments around it with respect to tephritids in California. I also agree that the field would benefit from thinking a little differently about the question of establishment, and on the dangers of intuition and extrapolation- this caution is common to anyone who has obtained an unexpected result from a model they created only to realize that their informal mental sketch before creating the model was flawed. As such, this paper is a valuable contribution to formalizing our thinking.

Aloha from Hilo,
NC Manoukis

Reviewer #2 (Remarks to the Author):

The reviewed manuscript shows great improvement and clarifications. I would firmly suggest again to add the author descriptors of the cited species in the manuscript (please refer to introduction and discussion section).

In my opinion the present version of the manuscript deserves the publication.